# Personalized Nutritional Assessment and Intervention for Athletes: A Network Physiology Approach

**DOI:** 10.3390/nu17233657

**Published:** 2025-11-23

**Authors:** Ainhoa Prieto, Maria Antonia Lizarraga, Natàlia Balagué

**Affiliations:** 1Departament de Biología Cel.lular, de Fisiología i d’Immunología, Facultat de Biologia, Universitat de Barcelona (UB), 08028 Barcelona, Spain; aprietma133@alumnes.ub.edu; 2Departament Nutrició, Ciencies de l’Alimentació i Gastronomía, Universitat de Barcelona (UB), 08921 Barcelona, Spain; mlizarraga@ub.edu; 3Institut Nacional d’Educació Fisica de Catalunya (INEFC), Complex Systems in Sport Research Group, Universitat de Barcelona (UB), 08195 Barcelona, Spain

**Keywords:** athlete’s healthcare, network physiology of exercise, gut microbiota, circadian rhythms, omics, complex adaptive systems, eating habits, constraints-led approach, semi-structured interviews

## Abstract

Nutritional assessment and intervention in athletes, a central focus of sports medicine and healthcare, has increasingly shifted in recent years toward precision nutrition—an approach that individualizes dietary recommendations according to genetic profile, microbiome composition, lifestyle factors, and health status. Despite its promising potential, this approach faces significant limitations, including the challenge of integrating complex and dynamic interactions among multilevel indicators, and the relatively high costs associated with omics technologies. The aim of this paper is to propose a nutritional assessment and intervention model grounded in the Network Physiology of Exercise, an emerging scientific field that investigates the horizontal and vertical dynamic interactions among nested physiological levels and conceptualizes athletes as complex adaptive systems (CAS). The proposal integrates social, environmental, behavioral and psychobiological information, extracted particularly from semi-structured interviews based on CAS properties. Accordingly, the traditional dietary assessment tools are replaced by open and guided interviews that allow professionals and practitioners to co-construct meaningful insights and extract qualitative data through a reflexive thematic analysis. From a CAS perspective, the multidimensional and multi-timescale personal and environmental constrains affecting their eating behavior were integrated through a hierarchically nested organization. Eliciting the dynamics of emotional contexts, behavioral patterns, and psychophysiological states, the interviews become both a method of assessment and an intervention in itself.

## 1. Introduction

Nutrition plays a fundamental role in athletes’ health and performance, providing not only fuel for training and competition but also supporting recovery [1]. The evolution of sports nutrition has deep historical roots: in ancient Greece and Rome, athletes followed specialized diets to enhance performance, with evidence of dietary strategies tailored to different sports disciplines [2].

Modern sports nutrition gained traction in the 20th century with significant advancements, particularly regarding the role of carbohydrate intake in endurance performance [3]. Despite such progress, the field has remained relatively static since the 2000s, emphasizing the need for a shift toward more personalized nutrition strategies. Because individual responses to nutritional interventions can vary substantially, recommendations have moved away from a one-size-fits-all approach toward tailored strategies that account for athletes’ unique characteristics [4].

Effective nutritional strategies begin with selecting appropriate assessment methods and techniques to identify nutrition-related problems, their causes, and their significance [5]. This process typically includes dietary evaluation [4], anthropometry and body composition analysis [6], biochemical testing [7], nutrition-focused clinical examination [4], and in-depth patient history [5,8]. Despite their utility, these methods often provide only a fragmented view of an athlete’s nutritional status, failing to capture the dynamic interactions among biopsychosocial factors and the multifaceted processes that influence performance, including training load, stress, sleep, and gut microbiome dynamics [7,9].

Fragmented and decontextualized assessment and intervention approaches become especially problematic in high-performance environments, where marginal gains can determine the difference between winning and injury, and between progress and burnout. In reality, athletes of all levels exist within a web of interactions that extend beyond diet and exercise. Factors like circadian rhythms, emotional stress, gut health, sleep quality, social pressure, and even belief systems co-regulate metabolism, performance, and recovery in dynamic ways. However, despite increasing recognition of these complexities [9], most assessment protocols still lack the structure and tools to integrate them meaningfully. In fact, what is missing is not more data, but a conceptual framework capable of integrating and interpreting this complexity in a coherent and athlete-centered way.

The Network Physiology of Exercise provides a framework that conceptualizes athletes as complex adaptive systems (CAS) interacting with their environment in dynamic and non-linear ways. Rather than treating physiological systems as isolated domains, Network Physiology examines how these systems co-regulate one another across contexts, contributing to both health and performance. The aim of this paper is to propose a nutritional assessment and intervention model grounded in the Network Physiology of Exercise, which views athletes as CAS.

## 2. Methods

The literature search was conducted primarily in the PubMed database using Boolean combinations of the following keywords: “nutritional assessment” AND “athletes”; “body composition” OR “dietary intake” OR “biochemical markers”; “energy expenditure” AND “assessment” AND “performance”; “clinical assessment” AND “elite athletes”; “omics” OR “gut microbiota” OR “precision nutrition”; “emotion regulation” OR “eating behavior” OR “interoception”; “sleep” OR “circadian rhythms” OR “chrono-nutrition”; and “network physiology” OR “constraints-led approach”.

The inclusion criteria were as follows: articles, reviews, and systematic reviews published in peer-reviewed journals (in English) between January 2010 and April 2025; studies focused on human male and female athletes; and studies describing methods, tools, or frameworks related to nutritional assessment. The exclusion criteria were: studies focused exclusively on clinical populations (e.g., diabetes, obesity), papers lacking methodological detail, and editorials, opinion pieces, or letters to the editor. Following the flexible approach typical of conceptual reviews, articles were screened based on their relevance to the topic. No formal appraisal or meta-analysis was conducted, as the aim was to provide an integrative and interpretative overview rather than to summarize all empirical findings.

## 3. Limitations of Gut Microbiome Analysis and Omics Technologies in Personalized Nutrition Assessments

Gut microbiome analysis and omics technologies seek to provide a detailed and individualized understanding of athletes’ physiological responses, adaptive capacity, and nutritional needs. However, while these methods offer exciting possibilities, their application has limitations in terms of feasibility, interpretation, and integration into practice.

The gut microbiome has emerged as a central modulator of host physiology and exemplifies the interaction among biopsychosocial factors in athletes’ nutrition [10,11,12]. Its composition and activity play crucial roles in nutrient metabolism, immune regulation, inflammation control, and cognitive and emotional function—all of which are key determinants of health and performance [13,14].

Microbiota diversity and metabolic activity are generally higher in athletes compared to sedentary individuals, with greater prevalence of beneficial strains such as Akkermansia muciniphila, Faecalibacterium prausnitzii, and Veillonella atypica—the latter has been shown to convert lactate into short-chain fatty acids (SCFAs), thereby supporting endurance and recovery [13,14]. These SCFAs, particularly butyrate, propionate, and acetate, enhance mitochondrial function, improve glucose metabolism, and reduce oxidative stress [15].

Beyond its role in metabolism, the gut microbiome contributes to the regulation of circadian rhythms [16], appetite hormones [17], and gene expression via epigenetic mechanisms influenced by microbial metabolites such as folate and B vitamins [18]. The bidirectional gut–brain axis further connects microbiome composition to mood, stress resilience, and interoceptive accuracy, thereby influencing emotional regulation and eating behaviors [13,19].

The gut microbiome is highly sensitive to environmental and lifestyle factors, including training load, stress, diet, illness, and travel [20]. This sensitivity makes it both a valuable biosensor of an athlete’s health and a fragile system prone to dysbiosis. Increased intestinal permeability (“leaky gut”) resulting from physical exertion can impair nutrient absorption and recovery while promoting inflammation [21]. Consequently, although gut microbiome analysis holds diagnostic and therapeutic potential, it requires careful interpretation and longitudinal monitoring to distinguish meaningful changes from transient fluctuations. To enhance validity and reproducibility, an international consensus has recently outlined standardized criteria for microbiota testing in clinical settings [22].

From its side, omics technologies enable the comprehensive analysis of biological systems at the molecular level—genomic (genes), transcriptomic (gene expression), proteomic (proteins), metabolomic (metabolites), and microbiomic (gut flora). These tools provide a multidimensional profile of the athlete’s internal environment, including energy availability, oxidative stress, recovery status, inflammation, muscle damage, and adaptation or recovery needs [23,24]. Consequently, they allow practitioners to move beyond generic dietary guidelines toward highly personalized nutrition strategies. When combined with wearable technology and artificial intelligence, integrative multi-omics frameworks have the potential to deliver real-time, adaptive insights to support decision-making in training and nutrition planning [25].

However, as pointed out by Network Physiology of Exercise advocates, sport-related phenomena cannot be understood if reduced to the collective characterization and quantification of pools of biological molecules, static representations of genomic and proteomic associations, and non-dynamic bottom-up group-pooled statistical inferences [26,27]. In addition, cost, limited access to equipment, variability in laboratory protocols, and difficulties in translating biomarkers into actionable recommendations remain significant hurdles of omics technologies [24,28]. Consequently, most omics-driven protocols remain confined to research or high-budget programs.

## 4. Personalized Nutrition Assessment and Intervention Model from a Network Physiology of Exercise Approach

To address the current gaps in personalized nutrition assessment and intervention, a new model grounded in the Network Physiology of Exercise is proposed [26,27,29,30]. When applied to nutrition, this framework allows practitioners to move beyond linear cause–effect thinking, embracing instead a systemic and dynamic approach that integrates multiple layers of information over time. This marks a departure from static assessment toward a more adaptive, personalized strategy for optimizing both health and performance in athletes.

The model recognizes that an athlete’s nutritional state is not static but emerges from the interplay of social, environmental, behavioral, and psychophysiological factors shaped by ongoing interactions with the environment. Rather than relying solely on point-in-time measurements or average-based recommendations, it promotes an integrated framework that reflects the fluid and context-dependent nature of CAS. This approach is particularly, though not exclusively, well suited to elite and endurance athletes, whose physiological systems are continuously responding to training load, travel, emotional fluctuations, recovery demands, and other stressors.

### 4.1. The Three-Layer Model

The proposed integrated, dynamic, and multi-layered model, grounded in the properties of complex adaptive systems (CAS), comprises Environmental, Behavioral, and Psychobiological layers (Figure 1). The integration of these layers provides a novel framework for personalizing the nutritional assessment and intervention of athletes.

The Environmental Layer encompasses the athlete’s cultural background, values and beliefs, social support systems, food environment, and lived experience. Rather than treating nutrition solely as a matter of willpower or education, this layer recognizes the deep influence of internal states (like interoception, motivation, or mood) and external conditions (travel, team culture, social media, or socioeconomic factors) on food behavior. Understanding how these behavioral dynamics evolve through different phases of training or competition is essential for sustaining long-term adherence and nutritional coherence. Importantly, such dynamic information cannot be revealed through gut microbiome analysis.The Behavioral Layer examines the dynamics of individual eating behavior. Athletes often encounter behavioral nutrition challenges that can compromise both health and performance. The most prevalent disordered eating behaviors include restrictive diets and binge-eating patterns, which may lead to Relative Energy Deficiency in Sport (RED-S)—a condition characterized by insufficient energy intake relative to expenditure, with consequences for metabolic, reproductive, and musculoskeletal health [31].

Erratic meal timing and inadequate post-training or competition nutrition can impair glycogen resynthesis, muscle repair, and subsequent performance [28]. Conversely, an overreliance on dietary supplements or unregulated ergogenic aids, coupled with poor hydration strategies—often reflecting behavioral misconceptions and limited nutrition literacy—can result in performance decrements and clinical complications. These behavioral nutrition challenges underscore the importance of comprehensive education, personalized interventions, and interdisciplinary support systems to promote both sustainable performance and long-term health.

The Psychobiological Layer focuses on emotional regulation, psychological stress, sleep quality, circadian alignment, immune status, gut health, endocrine responses, and other physiological systems that mediate nutritional requirements and adaptations. This layer captures the dynamic processes that affect energy regulation, recovery, and metabolic flexibility. For instance, poor sleep or psychological stress can lead to increased inflammatory markers and hormonal dysregulation, which in turn affect nutrient partitioning and absorption, appetite signals, gut function, and performance recovery [32,33]. However, traditional assessments rarely monitor these mediators in real-time, nor do they consider their co-regulation across systems. Tools such as actigraphy, heart rate variability, or subjective wellness questionnaires can provide valuable insight when interpreted within a broader systems perspective.

In this layer, omic technologies such as genomics, metabolomics, proteomics, microbiomics and transcriptomics, capturing some individual traits, can contribute to understand how athletes metabolize nutrients, respond to training, or recover from stress. For example, variations in caffeine metabolism genes can influence ergogenic effects or sleep disruption [34,35,36]; microbiome composition may shape energy harvest, immune function, or even emotional resilience [19,37]. While these technologies are still emerging in sports practice, their integration and contextualization within the broader behavioral and psychobiological layers is central to personalized nutrition strategies.

Inspired by CAS properties, the three nested layers (Figure 1) interact among them through circular causality. For instance, the environmental level (e.g., economic pressures such as contract negotiations, club expectations, comparisons with teammates, and/or media demands) affects the behavioral level (e.g., lack of restful sleep, drug use, emotional eating, sugar overconsumption, and altered eating patterns), which in turn affects the psychobiological level (e.g., increased cortisol levels, elevated inflammatory markers, gut microbiota disruption, and glucose fluctuations).

Furthermore, the interaction among the biopsychosocial constraints affecting the eating behavior can be hierarchically organized according to their relevant time scale (see Figure 2) [26,38]. Although the represented variables can fluctuate across multiple timescales, some of them are particularly relevant for personalized nutrition strategies. For example, fatigue may accumulate and persist for months, or it may arise as the consequence of a single training session (lasting only hours or days, as shown in Figure 2). Variables that change slowly, such as social habits and the food environment (upper-level variables), are more stable over time and tend to influence those that change more rapidly (lower-level variables), such as motivation or fear to eat certain types of food. In turn, through bottom-up circular causality among these variables, the motivation to eat specific foods can contribute to the formation of social habits. Similarly, emotional or mood changes alter physiological signaling and eating behavior, which in turn affect both physiological signaling and emotional states. A cascade of multilevel events can produce changes in athletes’ eating patterns. For instance, increased social expectations can lead to fear of failure, emotional distress, fatigue symptoms, altered physiological signaling (e.g., increased proinflammatory markers), and disordered eating behaviors (e.g., snacking, irregular schedules, and choices driven by stress and tiredness). In turn, healthy eating behaviors can help regulate physiological signaling, reduce fatigue symptoms, and improve emotional states and motivation.

Athletes’ nutrition is shaped by the nested organization of multiple environmental, behavioral, and psychobiological constraints operating across different timescales. Because these constraints are interrelated, interventions at one level can influence all others. In particular, targeting the social and environmental level may be especially effective due to its greater stability, potentially reducing the need for multiple simultaneous interventions. In conclusion, understanding the interrelatedness and integration of these constraints through the proposed nested model is essential for the effectiveness of nutritional assessment and intervention strategies.

### 4.2. Interactive Interviewing as Assessment and Intervention Tool for the Three-Layer Model

The practical application of the three-layer model for nutritional assessment and intervention requires not only comprehensive information but also appropriate tools to capture it. Given the need to consider multiple, multidimensional variables interacting in a nonlinear manner, as well as diverse outcomes emerging from individual dynamic contexts, the assessment tool must be flexible enough to facilitate the co-construction of rich and emergent narratives. These requirements make interactive interviewing the most suitable tool for both assessment and intervention [39].

Interactive interviews are focused on what interviewers and athletes can learn from the interview, as well as on the new information created by such interactions. In line with a dynamic complex systems perspective, it differs from many other forms of dietary evaluation, particularly food records, diaries, and standardized surveys [39]. As it allows sharing multidimensional personal and social experiences, the interaction often yields rich and unexpected information within the context of a developing relationship between interviewer and interviewed. The shared knowledge is further promoted because interactive interviews are flexible, enabling interviewers to pose unplanned questions as conversations unfold and purposeful curiosity arises, thereby generating novel insights in the process.

The combination of interactive and semi-structured interviews [40] provides space to guide the conversation toward topics that are meaningful from the perspectives of dynamic complex systems and the Network Physiology of Exercise, and more specifically, toward the interactions described in the three-layer model. In this way, interviewers and athletes can explore avenues of understanding that may not have been previously considered and actively co-construct relevant knowledge. Rich, detailed descriptions of experiences, coupled with the sharing of perspectives and interpretations, can yield nuanced insights into athletes’ decisions, values, motivations, beliefs, perceptions, feelings, and emotions—all of which profoundly influence eating behavior—and can help reveal the sociocultural dynamics underlying these behaviors.

Moreover, the hybrid combination of interactive and semi-structured interviews offers an opportunity to situate the dialogues within a social context and to focus on the interactive dynamics of the multidimensional factors influencing nutrition [41], effectively integrating assessment and intervention. It represents an excellent means of suggesting recommendations, correct unfounded nutrition beliefs or fears, and educating interviewees about the proposed model. Even the interview itself can be a topic for the analysis of interaction and talk, revealing both what is explicitly communicated and what remains unsaid.

Through targeted questioning, the interviewer guides athletes in uncovering the relationships among the multiple factors interacting within the three-layer model and helps them identify their personal idiosyncrasies. Questions such as “Which beliefs influence your motivation to eat certain types of food?”, “Is there any type of food that affects your mood state?”, or “Do you change your eating habits when experiencing particular mood states?” enable a deeper exploration of the individual triggers shaping nutritional status. By facilitating this level of personal insight, interactive interviewing can enhance the understanding of nutritional issues and contribute to truly personalized assessment and intervention strategies.

Although interactive interviewing aligns well with the goal of capturing emergent and co-constructed meaning, its strengths (e.g., reflexivity, adaptability, and dialogic depth) also create vulnerabilities related to reproducibility and ethical clarity. The interviewer’s own experiences, values, tone, and reactions can shape the dialogue, steering narratives in particular directions and influencing what ultimately emerges. Consequently, interviewers must possess strong reflexive skills and continuously examine how their presence affects the outcomes [31]. Moreover, building trust and reciprocity often requires time and relies on advanced interpersonal and facilitation skills, which are developed through deep understanding, formal training, and extensive practice [30]. Thus, interactive interviewing cannot be regarded as a simple task or a mechanical method that anyone can perform effectively after only a few attempts. Observing best-practice examples, receiving feedback on one’s performance, and engaging in ongoing training are essential. Ethical considerations are also critical, including ensuring appropriate support for both interviewers and interviewees following in-depth interviews [32].

In research contexts, interactive interviews require advanced qualitative methodological skills (e.g., discourse or narrative analysis) and offer limited generalizability. Therefore, they are most appropriate in contexts where understanding complexity and emergence is prioritized over standardization.

### 4.3. Guiding Athletes from Dependency to Self-Responsibility and Self-Efficacy to Warrant Personalized Nutrition

Although modern assessment systems seek to provide the basis for personalized nutritional recommendations, there is no real personalization without the active participation of athletes. Holistic, multifactorial, and dynamically changing dimensions can only be adequately integrated through co-designed (e.g., athletes-nutritionists) strategies that promote athlete’s self-responsibility and self-regulation.

One size fits all nutritional strategies are not just invalid for different persons, they are also invalid for the same individual if they are not adjusted to the changing environmental and psychobiological needs that may change at very short timescales. Transiting from dependency on professional prescriptions to self-efficacy requires an education of professionals on CAS properties [26]. The emerging criteria from co-designed nutrition strategies are warranted to foster adherence and develop interoceptive awareness, self-responsibility and self-regulation, essential to ensure healthy nutrition.

In summary, athlete’s nutrition should be conceptualized as a multidimensional phenomenon rather than reduced to the mechanistic management of calories and nutrients. The reductionist paradigm neglects the intricate, dynamic, and adaptive processes through which athletes nourish and regulate energy, recovery, and performance. Instead, nutrition should be understood as an evolving domain, continuously shaped by contextual demands, environmental contingencies, and individual variability.

Such a perspective necessitates recognition of the epistemological limitations inherent in the dominant body of scientific evidence, much of which emerges from experimental designs privileging control and standardization. While these contributions remain valuable, their translation into practice requires a contextualized and reflexive application that acknowledges athletes as CAS.

Accordingly, nutritional strategies should be developed through processes of co-design and co-adaptation, wherein both professionals and athletes actively cooperate to find adequate adjustments in response to personal, situational, and environmental dynamics. Advancing this paradigm requires education on CAS properties and athlete’s self-awareness, interoceptive sensitivity, and self-responsibility. In this way, athletes can reduce the potential adverse effects of rigid and prescriptive assessment and intervention approaches.

## 5. Concluding Remarks

The proposed nutritional assessment and intervention model, grounded in complex systems theory and the Network Physiology of Exercise, seeks to address limitations of current approaches to personalized nutrition by conceptualizing athletes as CAS. Rather than relying primarily on microbiota analyses or omics techniques, the model emphasizes the exploration of dynamic interactions across three interrelated layers that operate at different time scales: (a) social and environmental, (b) behavioral, and (c) psychophysiological. The multiple interconnected variables within these layers may be hierarchically organized according to their temporal dynamics. This dynamic dimension can be examined and integrated through interactive interviews. As interviews function as experiential processes, nutritional assessment and intervention become inherently interconnected. In this way, athletes, nutritionists, and health professionals collaboratively discover new insights and co-construct personalized and meaningful intervention strategies. Moreover, by fostering a deeper understanding of the athlete’s nutritional behavior, interactive interviews help develop essential competencies for effective nutritional management, including self-knowledge, self-responsibility, and self-efficacy. The implementation of this nutritional assessment and intervention model requires the adequate training of nutritionists and health professionals in complex systems theory and the principles underlying CAS. Future research is warranted to adapt and validate the model in various sporting contexts.

## Figures and Tables

**Figure 1 nutrients-17-03657-f001:**
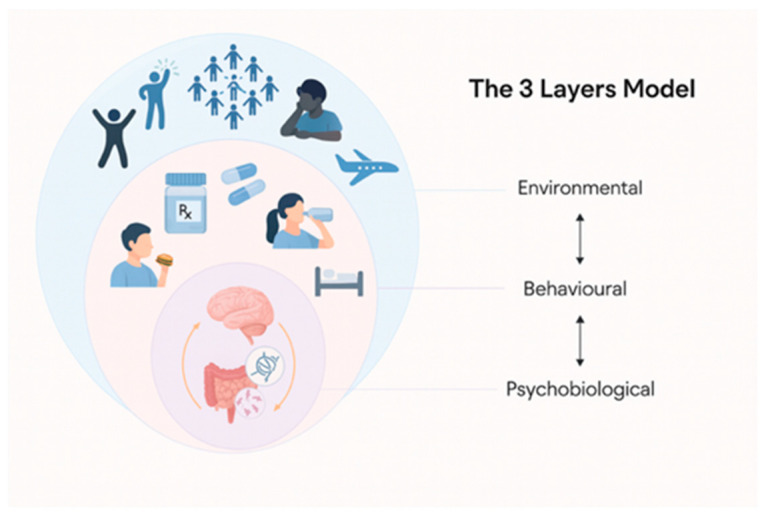
The three-layer nutritional assessment and intervention model. The double-headed arrows show the circular causality relationship among the environmental, behavioral and psychobiological levels.

**Figure 2 nutrients-17-03657-f002:**
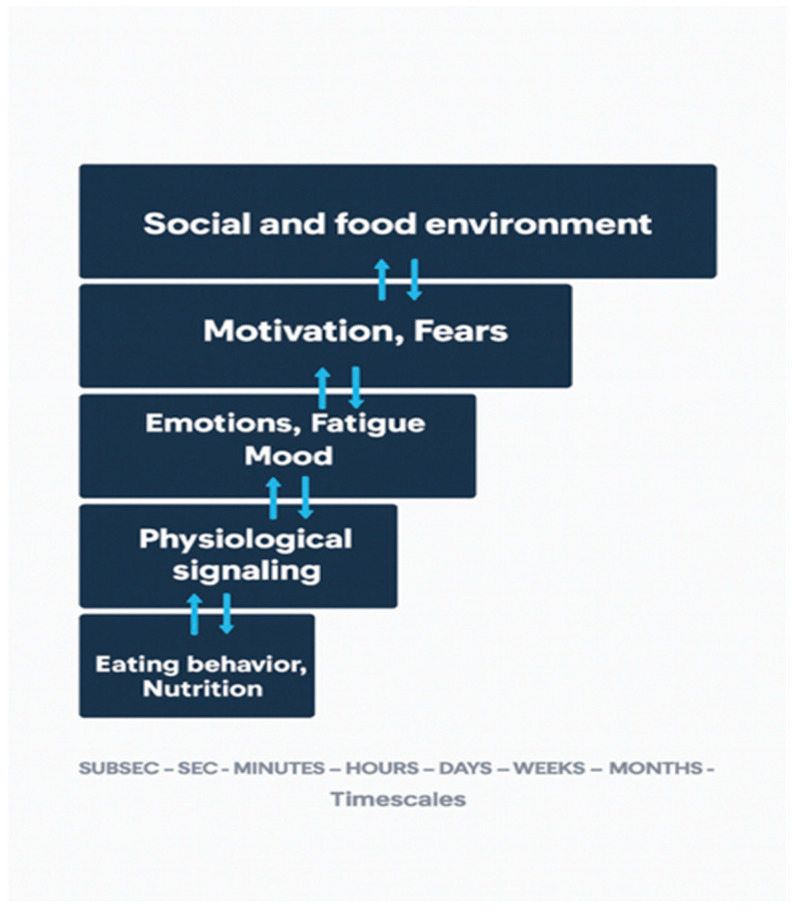
Nested organization of constraints affecting eating behavior. The double arrows show the circular causality relationship among the constraints. The timescales displayed in the horizontal axis are considered particularly relevant for personalized nutrition strategies. The upper level constraints (e.g., social environment) change slowly compared to the lower level ones (e.g., physiological signaling).

## Data Availability

No new data were created or analyzed in this study. Data sharing is not applicable to this article.

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
