# Peer review of "Personalized Nutritional Assessment and Intervention for Athletes: A Network Physiology Approach"

_nutrients, 2025, doi:10.3390/nu17233657_

Round 1

Reviewer 1 Report

Comments and Suggestions for Authors

The article is interesting, but purely theoretical. As such, it may serve as an introduction to conducting extensive practical research that will cover various methods of nutritional assessment along with their correlation with the results achieved. Furthermore, such research should be conducted in various sports disciplines, as it may turn out that certain assessment methods are not optimal in every case. 

A brief discussion in this case seems insufficient and is similar in form to the conclusions. In the discussion, the authors should cite various methods already in practical use, along with numerical results and their relationship to sports performance. The authors do not specify exactly how the tests will be conducted to ensure that they are repeatable and comparable in every case and discipline. The described assessment method is quite complicated in practice (rather for scientific research), and it would probably be necessary to develop a simpler model for everyday use by athletes, nutritionists, and coaches. The methodology of nutritional assessment is intended to improve diet planning, not to be evaluated itself. 

Author Response

Reviewer 1 Reply

R- Thank you for your positive review and constructive feedback. Your comments and suggestions helped us to significantly improve the clarity, quality and rigor of the present work. Please find below a detailed, point-by-point response to the questions. All revisions made to the manuscript have been marked in red for easy identification.

The article is interesting, but purely theoretical. As such, it may serve as an introduction to conducting extensive practical research that will cover various methods of nutritional assessment along with their correlation with the results achieved. Furthermore, such research should be conducted in various sports disciplines, as it may turn out that certain assessment methods are not optimal in every case.

R- The reviewer is right. The paper seeks to introduce a Network Physiology–based approach to nutritional assessment and intervention intending to enhance the understanding of athletes’ nutritional issues and to encourage practical research aimed at validating the proposed assessment and intervention tool. As mentioned in Section 4.3, the proposed tool requires education of interviewers on CAS properties as well as athlete’s self-awareness, interoceptive sensitivity, and sense of self-responsibility—qualities that are developed through practice with the assessment and intervention model. In our view, all types of sport specialties and population can benefit from it if appropriately adapted.

A brief discussion in this case seems insufficient and is similar in form to the conclusions. In the discussion, the authors should cite various methods already in practical use, along with numerical results and their relationship to sports performance. The authors do not specify exactly how the tests will be conducted to ensure that they are repeatable and comparable in every case and discipline. The described assessment method is quite complicated in practice (rather for scientific research), and it would probably be necessary to develop a simpler model for everyday use by athletes, nutritionists, and coaches. The methodology of nutritional assessment is intended to improve diet planning, not to be evaluated itself.

R- The discussion has been replaced by a concluding remarks section. The proposed assessment and intervention method is discussed in section 4.2. “Interactive interviewing as assessment and intervention tool of the three layers model”. Their advantages, stemming from the flexibility to consider numerous multidimensional variables interacting non-linearly within dynamic personal contexts, are also regarded as their primary vulnerabilities with respect to reproducibility and subjectivity. Research is warranted to adapt and validate it in various sport contexts and populations.

Reviewer 2 Report

Comments and Suggestions for Authors

The manuscript presents a conceptual review proposing a Network Physiology of Exercise–based framework for personalized nutrition in athletes. It is clearly written, original in its systems-level perspective, and well referenced with up-to-date literature (including 2024–2025 sources). The integration of behavioral, environmental, and psychobiological layers is coherent and innovative.

However, the paper is more conceptual than practical, and several issues could improve its clarity and utility for readers:

  1. Practical application – The model is described elegantly but remains theoretical. Please provide at least one concrete example or pilot scenario showing how the “interactive interview” or the three-layer model could be implemented in real practice (e.g., case vignette).

  2. Figure clarity – Figures 1–2 should be redrawn with better readability (fonts and labels), and each caption should clearly describe what the arrows or time scales represent.

  3. Structure – Sections 3.1–3.3 could benefit from clearer transitions; currently, the flow between “layers” and “interviewing” is abrupt.

  4. Language polishing – Minor grammar and phrasing issues (e.g., article use, verb tense consistency).

  5. Limitations – Add a short paragraph explicitly stating current limitations (lack of empirical validation, qualitative bias) and future directions (integration with digital tools, validation studies).

Overall assessment: The article is scientifically sound, well grounded in recent literature, and fits the scope of Nutrients. It only requires minor revisions for clarity and structure.

Recommendation: Minor revision (accept after minor edits).

Comments on the Quality of English Language

Minor grammar and phrasing issues (e.g., article use, verb tense consistency).

Author Response

Reply to Reviewer 2

R- Thank you for your positive review and constructive feedback. Your comments and suggestions helped us to significantly improve the clarity, quality and rigor of the present work. Please find below a detailed, point-by-point response to the questions. All revisions made to the manuscript have been marked in red for easy identification.

It seems there may be a typographical oversight in the manuscript title, where the word “title” appears to have been inadvertently included. I would suggest removing the word “title” for greater clarity and consistency.

R- Done

INTRODUCTION

I would like to thank the authors for introducing a relevant and interesting topic and for providing a clear historical overview that effectively highlights the field's growing importance.

However, in lines 43–45, the word “advancements” appears twice in proximity, making the sentence sound slightly repetitive. I would suggest replacing one occurrence with a suitable synonym (e.g., “developments,” “progress,” or “innovations”) to improve the stylistic flow and overall readability of the paragraph.

R- Thank you for the correction. The terms have been rewritten accordingly.

In addition, I suggest including a brief “Materials and Methods” section to explain how the literature search was conducted. This would enhance the transparency and reproducibility of the review process.

R- A methods section has been added after the introduction.

  1. GUT MICROBIOME

I would like to thank the authors for providing an insightful overview of a highly relevant topic and for effectively linking the concepts of nutrition and, increasingly, personalized approaches. However, considering the current scientific landscape, in which the study of the gut microbiota is becoming an increasingly tangible reality, I suggest including some discussion or references regarding which phyla, or preferably genera and species, have been most extensively studied in the context of nutrition and personalized nutrition in athletes. This would help lay the groundwork for the concept of personalization itself.

R- A paragraph together with several references have been included regarding genera and species: Microbiota diversity and metabolic activity tend to be higher in athletes compared to sedentary individuals, with greater prevalence of beneficial strains such as Akkermansia muciniphila, Faecalibacterium prausnitzii, and Veillonella atypica, the latter shown to convert lactate into short-chain fatty acids (SCFAs), thereby contributing to endurance and recovery [13, 14]. These SCFAs, especially butyrate, propionate, and acetate, enhance mitochondrial function, improve glucose metabolism, and reduce oxidative stress (Barton et al., 2018).

The following references connecting microbiota with personalized nutrition in athletes have been added to section 3 and the reference list

Fontana A, Porcar M, Berg G, et al. The human gut microbiome of athletes: metagenomic and metabolic insights. Microbiome. 2023;11(1):161.

Nolte F, Werner J, Fricker N, et al. Optimizing the gut microbiota for individualized performance in athletes. Nutrients. 2023;15(2):356.

Chen Y, Liao J, Liu X, et al. Dietary patterns, gut microbiota and sports performance in athletes. Nutrients. 2024;16(11):1634.

Boisseau N, Rufo C, Pelsy F, et al. The nutrition–microbiota–physical activity triad. Nutrients. 2022;14(5):1050.

Moreover, although it is true that microbiota research still suffers from considerable heterogeneity in the analytical methods employed—perhaps due to the methodological “anarchy” of past years—it is worth emphasizing that a recent international consensus has been published. This consensus outlines standardized criteria that microbiota analysis methods should follow to be considered validated and reproducible tools. Mentioning this would strengthen the manuscript and further highlight how the study of the gut microbiota is evolving into an increasingly established and structured field.

R- A short sentence has been included at the end of section 3 introducing the following citation: Porcari and col. International consensus statement on microbiome testing in clinical practice. Lancet Gastroenterol Hepatol. 2025 Feb;10(2):154-167. doi: 10.1016/S2468-1253(24)00311-X.

  1. I would like to thank the authors for introducing the innovative concept of applying the Network Physiology of Exercise framework to personalized nutrition, which adds an interesting theoretical dimension to the paper. However, this section would strongly benefit from the inclusion of supporting references to enhance the scientific robustness and contextual grounding of the proposed model.

R- The following references have been added:

Bashan, A. et al. Network physiology reveals relations between network topology and physiological function. Nat Commun. 3, 702. https://doi.org/10.1038/ncomms1705 (2012),

Ivanov, P. C. The new field of network physiology: Building the human physiolome. Front Netw Physiol. 1, 711778. h t t p s : / / d o i . o r g / 1 0 . 3 3 8 9 / f n e t p . 2 0 2 1 . 7 1 1 7 7 8 (2021).

Balagué N et al. (2020) Network Physiology of Exercise: Vision and Perspectives. Front. Physiol. 11:611550. doi: 10.3389/fphys.2020.611550

Balagué N, et al. Network Physiology of Exercise: Beyond Molecular and Omics Perspectives. Sports Med Open. 2022 Sep 23;8(1):119. doi: 10.1186/s40798-022-00512-0

3.1. The three-layer model

Thanks for clearly outlining the potential of omics technologies in understanding individual variability among athletes. However, I would recommend adding appropriate literature references to support the statements provided, particularly the one regarding caffeine metabolism gene variants and their impact on ergogenic effects and sleep disruption. Including such references would strengthen the scientific grounding of this section and improve its credibility.

R- The following references have been added:

Barreto G, Grecco B, Merola P, Reis CEG, Gualano B, Saunders B. Novel insights on caffeine supplementation, CYP1A2 genotype, physio-logical responses and exercise performance. Eur J Appl Physiol 2021;121:749–69. https://doi.org/10.1007/s00421-020-04571-7.

Rahimi M, Semenova E, John G, Fallah F, Larin A, Generozov E, et al. Effect of ADORA2A Gene Polymorphism and Acute Caffeine Sup-plementation on Hormonal Response to Resistance Exercise: A Dou-ble-Blind, Crossover, Placebo-Controlled Study. Nutrients 2024;16:1803. https://doi.org/10.3390/nu16121803.

Reichert CF, Deboer T, Landolt H. Adenosine, caffeine, and sleep–wake regulation: state of the science and perspectives. Journal of Sleep Research 2022;31:e13597. https://doi.org/10.1111/jsr.13597

Cryan JF, Dinan TG. Mind-altering microorganisms: the impact of the gut microbiota on brain and behaviour. Nat Rev Neurosci. 2012 Oct;13(10):701-12. doi: 10.1038/nrn3346.

Belkaid Y, Hand TW. Role of the microbiota in immunity and inflammation. Cell. 2014 Mar 27;157(1):121-41. doi: 10.1016/j.cell.2014.03.011

I would like to thank the authors for presenting a comprehensive and well-articulated conceptual framework describing the multilevel and dynamic interactions influencing athletes’ nutrition. However, this section would benefit from the inclusion of appropriate literature references to support the claims, particularly regarding the effects of slow-changing variables on faster-changing variables, as well as the bidirectional influences between emotional states, physiological signaling, and eating behavior. Adding citations would strengthen the scientific foundation of this model and guide readers to relevant supporting studies.

R- The following references to support the claims have been added:

Balagué N et al. (2020) Network Physiology of Exercise: Vision and Perspectives. Front. Physiol. 11:611550. doi: 10.3389/fphys.2020.611550

Balagué, N., Pol, R., Torrents, C. et al. On the Relatedness and Nestedness of Constraints. Sports Med - Open 5, 6 (2019). https://doi.org/10.1186/s40798-019-0178-z

Some examples have been also provided in section 4.1. to clarify the proposed model.

3.2, 3.3, 4

I would like to commend the authors for producing a well-structured and clearly presented manuscript. The conceptual framework and narrative are coherent and thoughtfully organized. However, the work would benefit from the inclusion of additional bibliographic references to support key statements and claims throughout the text. Furthermore, I would suggest adding a section on the limitations of the study, which would provide a more balanced discussion and help contextualize the findings for the reader.

R- A specific paragraph about limitations of the proposal has been added to section 4.2. Interactive interviewing as assessment and intervention tool of the three layers model. Future research directions are recommended in the concluding remarks section.

Reviewer 3 Report

Comments and Suggestions for Authors

It seems there may be a typographical oversight in the manuscript title, where the word “title” appears to have been inadvertently included. I would suggest removing the word “title” for greater clarity and consistency.
INTRODUCTION
I would like to thank the authors for introducing a relevant and interesting topic and for providing a clear historical overview that effectively highlights the field's growing importance.
However, in lines 43–45, the word “advancements” appears twice in proximity, making the sentence sound slightly repetitive. I would suggest replacing one occurrence with a suitable synonym (e.g., “developments,” “progress,” or “innovations”) to improve the stylistic flow and overall readability of the paragraph.

In addition, I suggest including a brief “Materials and Methods” section to explain how the literature search was conducted. This would enhance the transparency and reproducibility of the review process.

2. GUT MICROBIOME
I would like to thank the authors for providing an insightful overview of a highly relevant topic and for effectively linking the concepts of nutrition and, increasingly, personalized approaches. However, considering the current scientific landscape, in which the study of the gut microbiota is becoming an increasingly tangible reality, I suggest including some discussion or references regarding which phyla, or preferably genera and species, have been most extensively studied in the context of nutrition and personalized nutrition in athletes. This would help lay the groundwork for the concept of personalization itself.

Moreover, although it is true that microbiota research still suffers from considerable heterogeneity in the analytical methods employed—perhaps due to the methodological “anarchy” of past years—it is worth emphasizing that a recent international consensus has been published. This consensus outlines standardized criteria that microbiota analysis methods should follow to be considered validated and reproducible tools. Mentioning this would strengthen the manuscript and further highlight how the study of the gut microbiota is evolving into an increasingly established and structured field.

3.
I would like to thank the authors for introducing the innovative concept of applying the Network Physiology of Exercise framework to personalized nutrition, which adds an interesting theoretical dimension to the paper. However, this section would strongly benefit from the inclusion of supporting references to enhance the scientific robustness and contextual grounding of the proposed model.

3.1. The three-layer model
Thanks for clearly outlining the potential of omics technologies in understanding individual variability among athletes. However, I would recommend adding appropriate literature references to support the statements provided, particularly the one regarding caffeine metabolism gene variants and their impact on ergogenic effects and sleep disruption. Including such references would strengthen the scientific grounding of this section and improve its credibility.

I would like to thank the authors for presenting a comprehensive and well-articulated conceptual framework describing the multilevel and dynamic interactions influencing athletes’ nutrition. However, this section would benefit from the inclusion of appropriate literature references to support the claims, particularly regarding the effects of slow-changing variables on faster-changing variables, as well as the bidirectional influences between emotional states, physiological signaling, and eating behavior. Adding citations would strengthen the scientific foundation of this model and guide readers to relevant supporting studies.

3.2, 3.3, 4
I would like to commend the authors for producing a well-structured and clearly presented manuscript. The conceptual framework and narrative are coherent and thoughtfully organized. However, the work would benefit from the inclusion of additional bibliographic references to support key statements and claims throughout the text. Furthermore, I would suggest adding a section on the limitations of the study, which would provide a more balanced discussion and help contextualize the findings for the reader.

Author Response

Reply to Reviewer 3

R- Thank you for your positive review and constructive feedback. Your comments and suggestions helped us to significantly improve the clarity, quality and rigor of the present work. Please find below a detailed, point-by-point response to the questions. All revisions made to the manuscript have been marked in red for easy identification.

The manuscript presents a conceptual review proposing a Network Physiology of Exercise–based framework for personalized nutrition in athletes. It is clearly written, original in its systems-level perspective, and well referenced with up-to-date literature (including 2024–2025 sources). The integration of behavioral, environmental, and psychobiological layers is coherent and innovative.

However, the paper is more conceptual than practical, and several issues could improve its clarity and utility for readers:

Practical application – The model is described elegantly but remains theoretical. Please provide at least one concrete example or pilot scenario showing how the “interactive interview” or the three-layer model could be implemented in real practice (e.g., case vignette).

R- Two examples have been added to section 4: Personalized nutrition assessment and intervention model from a Network Physiology of Exercise approach.

To illustrate the relationship among the three layers, the following paragraph has been added: “The environmental level (e.g., economic pressures such as contract negotiations, club expectations, comparisons with teammates, and/or media demands) affects the behavioral level (e.g., lack of restful sleep, drug use, emotional eating, sugar overconsumption, and altered eating patterns), which in turn affects the psychobiological level (e.g., increased cortisol levels, elevated inflammatory markers, gut microbiota disruption, glucose fluctuations).

To illustrate Figure 2: A cascade of multilevel events can produce changes in athletes’ eating patterns. For instance, increased social expectations can lead to fear of failure, emotional distress, fatigue symptoms, altered physiological signaling (e.g., increased proinflammatory markers), and disordered eating behaviors (e.g., snacking, irregular schedules, and choices driven by stress and tiredness). In turn, healthy eating behaviors can help regulate physiological signaling, reduce fatigue symptoms, improve emotional states and motivation.

The practical aspects of the interactive interview are discussed in section 4.2

Figure clarity – Figures 1–2 should be redrawn with better readability (fonts and labels), and each caption should clearly describe what the arrows or time scales represent.

R- Figures have been revised, and their captions have been expanded to provide more detailed descriptions.

Structure – Sections 3.1–3.3 could benefit from clearer transitions; currently, the flow between “layers” and “interviewing” is abrupt.

R- A paragraph emphasizing the suitability of interactive interviews for assessing and intervening in CAS has been added to better connect both subsections. In addition, the section heading has been expanded to more clearly link the three-layers model with the interactive interviews: ‘Interactive interviewing as an assessment and intervention tool for the three-layers model’.” A paragraph emphasizing the adequacy of interactive interviews for assessing and intervening on CAS has been added to connect both subsections. The section heading has been enlarged to relate the three layers model with the interactive interviews “Interactive interviewing as assessment and intervention tool of the three layers model”.

Language polishing – Minor grammar and phrasing issues (e.g., article use, verb tense consistency).

R- The language has been revised.

Limitations – Add a short paragraph explicitly stating current limitations (lack of empirical validation, qualitative bias) and future directions (integration with digital tools, validation studies).

R- A limitations paragraph has been added to Section 4.2, and a future directions sentence has been incorporated into the concluding remarks section.

Overall assessment: The article is scientifically sound, well grounded in recent literature, and fits the scope of Nutrients. It only requires minor revisions for clarity and structure.

Recommendation: Minor revision (accept after minor edits).

Round 2

Reviewer 3 Report

Comments and Suggestions for Authors

No additional comments